# Engineered Peptides Enable Biomimetic Route for Collagen Intrafibrillar Mineralization

**DOI:** 10.3390/ijms24076355

**Published:** 2023-03-28

**Authors:** Aya K. Cloyd, Kyle Boone, Qiang Ye, Malcolm L. Snead, Paulette Spencer, Candan Tamerler

**Affiliations:** 1Bioengineering Program, University of Kansas, Lawrence, KS 66045, USA; 2Institute for Bioengineering Research, University of Kansas, Lawrence, KS 66045, USA; 3Department of Mechanical Engineering, University of Kansas, Lawrence, KS 66045, USA; 4Center for Craniofacial Molecular Biology, Herman Ostrow School of Dentistry of USC, University of Southern California, Los Angeles, CA 90007, USA

**Keywords:** dental caries, resin composites, collagen self-assembly, peptide design, collagen binding peptide, dentin adhesive interface, calcium phosphate mineralization

## Abstract

Overcoming the short lifespan of current dental adhesives remains a significant clinical need. Adhesives rely on formation of the hybrid layer to adhere to dentin and penetrate within collagen fibrils. However, the ability of adhesives to achieve complete enclosure of demineralized collagen fibrils is recognized as currently unattainable. We developed a peptide-based approach enabling collagen intrafibrillar mineralization and tested our hypothesis on a type-I collagen-based platform. Peptide design incorporated collagen-binding and remineralization-mediating properties using the domain structure conservation approach. The structural changes from representative members of different peptide clusters were generated for each functional domain. Common signatures associated with secondary structure features and the related changes in the functional domain were investigated by attenuated total reflectance Fourier-transform infrared (ATR-FTIR) and circular dichroism (CD) spectroscopy, respectively. Assembly and remineralization properties of the peptides on the collagen platforms were studied using atomic force microscopy (AFM). Mechanical properties of the collagen fibrils remineralized by the peptide assemblies was studied using PeakForce-Quantitative Nanomechanics (PF-QNM)-AFM. The engineered peptide was demonstrated to offer a promising route for collagen intrafibrillar remineralization. This approach offers a collagen platform to develop multifunctional strategies that combine different bioactive peptides, polymerizable peptide monomers, and adhesive formulations as steps towards improving the long-term prospects of composite resins.

## 1. Introduction

Polymeric restorative materials have revolutionized the treatment of dental caries by allowing clinicians a “one-of-a kind” in situ tissue engineering approach afforded by resin–dentin bonding [1,2,3,4,5]. Despite significant advances in composite resins for treating teeth, the low durability of the current dental adhesives continues to be a major health burden. The Global Oral Health Status 2022 report released by the World Health Organization (WHO) estimated 2 billion people worldwide suffer from caries in their permanent teeth [6,7]. The retention of composite resins relies on the adhesive system, which infiltrates into the collagen matrix to adhere to dentin [1,8,9,10,11]. Adhesion and adhesive performance on dentin are especially challenging due to the complexity of dentin, a mineralized dynamic biological composite tissue composed of 70% hydroxyapatite (HA), 30% type I collagen, multiple non-collagenous proteins, and water [12,13,14]. Due to the multi-scaled and multi-faceted events taking place at this complex resin–dentin adhesive interface, strategies developed so far have been slow to address the short lifespan of resin–dentin bonding and subsequently prevent the repeated failure of composite resin restorations [3,15]. Composite restorations will substantially benefit from the next generation of approaches based upon bioactive and biohybrid pathways that address the ongoing multifaceted events at this complex interface [2,15,16,17,18].

Resin–dentin bonding can be facilitated through acid-etching to remove the mineral phase which provides space to facilitate the infiltration of the adhesive system into the collagen matrix of dentin. Structurally homogeneous infiltration of the resin monomers into this complex matrix is difficult. Monomers are expected to fill the intra- and inter-fibrillar spaces within the collagen network and polymerize in situ to fully seal the region [4,10,19,20]. However, the resulting resin–dentin interdiffusion zone suffers from poor infiltration of resin monomers into the demineralized dentin matrix (collagen network). Penetration capacity of the monomers, caries pathophysiology, the expanded state of dentin collagen structure, and the variation on mineral content all affect the integrity of the resin–dentin interdiffusion zone. Unbound and entrapped water in the collagen network further accelerates the hydrolysis of resin and intensifies the vulnerability at this hybrid layer.

Despite the numerous approaches that have been developed to eliminate residual water from the collagen network, residual water is trapped within the hybrid layer. The residual water interferes with polymerization, plasticizes the adhesive and facilitates hydrolysis of the ester bonds in the methacrylate-based adhesives. The carboxylate and alcohol by-products of the ester hydrolysis are even more hydrophilic than the original adhesive network, thus increasing water sorption into the hybrid layer causing a cascade of degradation [4,8,10,11,21,22,23,24]. This self-perpetuating feed-forward cycle is aggravated by the insufficient infiltration of adhesive into the demineralized dentin collagen [25,26]. Consequently, the water-rich, highly porous collagen fibrils become exposed, allowing further sorption into the adhesive. Over time, degradation reaches a point where microleakage starts to occur, expediating the diffusion of oral fluids (containing e.g., salivary esterase) and bacteria (e.g., *S. mutans*) into the site [4,19,22,24,26,27,28]. These contaminants accelerate the degradation cascade of the adhesive, thus compromising its bond integrity, furthering tooth demineralization and caries progression at the site—resulting in the ultimate failure of the restoration [4,20,21,22,24]. It is recognized that the unprotected collagen fibers have a highly porous structure. Strategies that enable achieving better penetration and protecting the collagen network would be key to providing interfacial stability at the complex dentin/adhesive interface.

The dentin extracellular matrix contains collagens, non-collagenous proteins, and self-assembled collagen fibrils cooperating to stabilize and guide mineral growth. Collagen mineralization is considered to be mediated by interactions between negatively charged complexes of amorphous calcium phosphate (ACP) precursors and the collagen fibers. The ACP precursors are formed due to interactions between ionic components with the proteins controlling the mineral deposition and phase transformation precipitation. The ACP precursors penetrate the collagen fibrillar matrix and then transform into hydroxyapatite, resulting in the excellent mechanical properties observed in mineralized dentin. Developing a better understanding of the biomineralization processes and mechanisms has accelerated biomimetic design strategies for tissue repairs [15,29]. Non-collagenous proteins, composed mainly of acidic residues and their analogs, as well as peptides enriched in acidic amino acids were proposed to stabilize the mineralization and prevent degradation at the soft–hard tissue interfaces including the dentin/adhesive interface [30,31,32,33].

Our group has focused on approaches that mimic biological design of complex materials and interfaces. Using combinatorial and computational approaches, our group has been designing peptides with specific affinity for inorganic materials including calcium-phosphate minerals (Ca-P). Prior work with these peptides showed that they can induce calcium phosphate mineralization by exerting control over mineralization kinetics and mineral morphology, under specific conditions. We reported phage-display selection of a hydroxyapatite-binding peptide (HABP1: MLPHHGA) and reported on its mineralization properties in constrained and unconstrained forms [34]. The HABP peptide can be combined with another bioactive peptide or protein to generate novel biological agents. We also demonstrated that the resulting multifunctional properties of these peptides can be further improved using computational modeling and predictive tools. By designing HABP coupled to fluorescent proteins, we demonstrated that the resulting molecule can be used to label mineralized tissues [35], as well as direct remineralization on deficient dentin surfaces [4]. Furthermore, these peptides can be conjugated with a monomer and retain their bioactive property, even as they conform into a polymeric network in adhesive formulation [21]. Our recent studies include the formulation of a “bio-hybrid” adhesive, designed by conjugating a methacrylate monomer with HABP for remineralization or conjugated with an antimicrobial peptide to inhibit oral pathogens [21].

Intrafibrillar remineralization of collagen at the “hybrid layer” could be a critical pathway to address the vulnerability of the exposed demineralized dentin collagen and improve long-term repair of the tooth [36]. Recent approaches for the intrafibrillar remineralization with peptides involve using the casein phosphopeptide-amorphous calcium phosphate complex [37,38], beta-sheet self-assembled peptide hydrogels (P11-4 [39], ID8 [40], RAD/KLT [41]), amelogenin-derived peptide P26 [42], and mineralization-promoting peptides (MMP3 [43,44]) among others. These methods either focus on mineral formation at the site or involve stabilizing the collagen fibrils by generating a scaffold to achieve intrafibrillar mineralization [45]. While all these approaches could be promising, none of these approaches are specifically targeting collagen accompanied by mineral deposition taking place at collagen intrafibrillar and interfibrillar sites. A platform that enables the study of different molecular pathways while targeting collagen could contribute to the design of a hybrid layer with improved properties.

In this study, we propose an approach that builds upon an engineered peptide designed to specifically target collagen and guide mineral deposition at intrafibrillar collagen sites. Here, we advance a simple, dynamic, and flexible collagen self-assembly platform upon which to test the activity and efficacy of various designed peptides. By combining computational and experimental tools, a multi-functional peptide is designed to combine a hydroxyapatite-binding peptide (HABP1) with a collagen-binding peptide motif (TKKLTLRT) using a spacer sequence to minimize interdomain interactions. By combining experimental and computational approaches, we examined self-assembly and remineralization properties of the peptides on the collagen platform.

## 2. Results

Human dentin and rat tail type I collagen samples were investigated as part of our initial studies. Two-dimensional RAMAN light microscopy was used to identify the differences in chemical signature between demineralized and intact dentin on a human tooth specimen. Figure 1A shows the light micrograph representation of the two regions, with Figure 1B reflecting the spectral analysis. For intact dentin, the characteristic peaks associated with PO_4_^3−^ and CO_3_^2−^ are observed at 960 cm^−1^ and 1070 cm^−1^, respectively. The intensity of these mineral bands was decreased in the demineralized dentin, whereas the spectral signatures reflecting type I collagen became clearer: 1667 cm^−1^ (amide I), 1460 cm^−1^ (CH_2_), and the doublet observed from 1215 cm^−1^ to 1310 cm^−1^ (amide III) [3,46,47]. The chemical signatures observed in both intact and demineralized dentin were compared to spin-coated (SC) rat tail type 1 collagen on glass using RAMAN microscopy. To mimic biological systems, we adapted alkaline phosphatase (ALP)-mediated mineralization. As a key enzyme, ALP promotes mineralization by hydrolyzing the pyrophosphate and releasing inorganic phosphate. Raman spectra of the collagen samples were obtained with and without undergoing alkaline phosphatase-mediated mineralization (Figure 1C). The indicative collagen peaks observed in demineralized dentin and the mineral peaks seen with intact dentin were respectively observed in the SC collagen and the mineralized SC collagen. The amide peaks observed on the SC collagen samples were diminished in the mineralized SC collagen sample; however, the 1454–1460 cm^−1^ Raman spectral feature associated with CH_2_ related to collagen is notable.

We next evaluated the use of HABP1 (MLPHHGA) to potentially direct mineralization within the adhesive/demineralized dentin hybrid layer. To do this we used RAMAN Divisive Clustering Analysis (DCA), a hierarchical k-means clustering method which is useful for identifying the internal structure of multi-layer datasets. We mapped the most distinctive mineral composition variations associated with the collagen matrix structural variations according to their RAMAN spectra. Our mapping includes two groups, where HABP1 mineral group II shows a more mature mineral formed compared to that observed for HABP1 mineral group I, a distinction based on the narrower full-width half-maximum (FWHM) of the phosphate peak of mineral group I. Figure 2A shows 2D RAMAN spectroscopy with and without HABP1 incorporation into the adhesive/dentin specimen, following the mineralization procedure. Figure 2A shows that HABP1 mediates mineralization shown with a RAMAN peak consistent with CaPO_4_ formation (960 cm^−1^) which does not appear after the mineralization procedure without peptide. Figure 2B further analyzes the 2D Raman spectra, with pseudo-coloring being applied by the DCA method. The DCA method measures similarity in three parameters: ratio of 960/1460 peak (e.g., mineral-to-matrix ratio), ratio of 1078/960 (gradient mineral carbonation, GMC: carbonate to phosphate) peak, and ratio of the 960/1667 peaks (phosphate to collagen with retained amide I structure, Ca/P collagen). The spectral feature at 1460 cm^−1^ is assigned to the CH_2_ wag for collagen, and 1667 cm^−1^ is the amide I peak for the collagen used [48]. The 960 cm^−1^ peak is the ν1 phosphate stretching vibration associated with the mineral and 1078 cm^−1^ is the band for ν1 carbonate. The method clusters the following groups: collagen only, mineral group I, and mineral group II. This analysis method is summative, meaning that the mineral groups discovered are the mineral formation groups which are most distinctive. The distinction between the mineral groups is based on how much collagen is present at the location of the mineral, how much of the present collagen’s two noted amide peaks signature are maintained, and by how much carbonate is in the mineral.

Figure 2C shows that mineral group I exhibits greater relative intensity of mineral to amide I (phosphate, 960)/amide I, 1667), an increased ratio of carbonate to phosphate (1078/960), and a decreased mineral-to-matrix ratio (phosphate/CH_2_ wag, collagen). The value of 1667 cm^−1^ has previously been noted as a beta-sheet structure [48]. Mineral group II exhibits less carbonate compared to mineral group I. Due to the lower carbonate content observed and narrower FWHM, i.e., full width at half maximum, we propose that mineral group II is a more mature mineral formation, being closer to hydroxyapatite compared to mineral group I. Figure 2D shows where the two mineral groups, which are represented by the spectra on the left, appear in the isolated DCA maps. The top right panel shows the DCA map dominated by collagen, the top left panel shows the DCA map of mineral group 1, and the bottom left panel shows the DCA map of mineral group II. Comparing the class spectra to the two mineral groups on the left, we see that the mineral groups are not segregated exclusively by how much mineral absorbance occurs at a location. This shows that we can distinguish the activity of HABP1 with respect to the background collagen and produce mature biomimetic mineral. When co-assembled with collagen, it is plausible to expect this peptide to direct biomimetic intrafibrillar mineralization at this complex interface.

To extend the remineralization capability into the exposed collagen network along the demineralized dentin matrix, we designed a bifunctional chimeric peptide sequence combining collagen-binding and remineralization domains using a spacer sequence. Using PEP-FOLD3, we compared folded structures of the chimeric peptide sequence with the corresponding folded domains alone. Our motivation is that identifying structural change in the incorporated chimeric peptide domain relates to change in the function of the domain. The extend of these changes is explored through the comparisons of different spacers with varying flexibility and length.

The spacers fit into three general categories according to the inter-domain relationships: (1) inter-residue contacts, (2) globular conformation, and (3) bi-function accessibility conformation. The inter-residue contacts limit the conformational flexibility of the domains the most directly, while the globular conformation limits the accessibility of the domain surfaces through shared interfaces by those domains. We investigated helical motifs with low GRAVY scores (KGSVLSA, PKSALQEL) and high GRAVY scores (GLALLGWG, LGWLSAV, WLMNYFWPL, and YLMNYLLPY). The limitation of these helical motifs was to impose conformation where the active domains shared interfaces. The most promising conformation type for retaining independent functions is the bi-functional accessibility conformation because each domain has the most access to interact with its molecular recognition partner. Figure 3 provides a subset of selected peptide structures that are incorporated with three different spacer motifs. The GSGGG selected as short spacer sequence with high flexibility resulted in inter-domain contacts. The KGSVLSA was selected due to its low flexibility to develop a shared domain interface. Conversely, the APA was selected as a very short sequence with low flexibility which offered the greatest accessibility for the domains. We chose the APA spacer which reduces the probability of inter-domain interactions.

The spacer length and its backbone shape are critical for inter-domain residue interactions, if any, are likely to induce conformational shifts between the domain structures. In Figure 4, each of the single peptide domains (CBP Figure 4A–D, HABP1 Figure 4E–H) and the chimeric peptide (Figure 4I–L) are shown with their electronegativity surface features. The provided structures are energy-minimized folded structures chosen as a cluster representation by PEP-FOLD3 [49] and rendered by UCSF Chimera [50]. The functional domains of the chimeric peptide (Figure 4I–L) folded structure surfaces are orthogonal to each other in the structure views shown in Figure 4. It is notable that the domains do not form a hairpin-like structure in which the collagen-binding domain and the hydroxyapatite-binding domain contact residues with each other.

Based on the domain structure conservation approach analysis, we selected CBP-APA-HABP1 to test experimentally and we synthesized peptide. The secondary structures of the CBP, HABP1, and CBP-HABP1 were investigated using the FT-IR spectra in their lyophilized state (Figure 5). The amide I band relates to the stretching of the backbone carbonyl. CBP-HABP1 and CBP have their highest amide I band peak related to beta-sheet structures with peak absorbance at 1624 ± 1 cm^−1^ [48]. CBP-HABP1 and CBP also share an amide I peak of 1662 cm^−1^, which is related to α-helix formation. The dominant peak for lyophilized linear HABP1 is at 1646 cm^−1^, which is in the random assignment region [48]. We observed a main peak at 3280 cm^−1^ in the group A region (NH stretching) for both the CBP and HABP1 peptides, which was observed to be shifted to 3270 cm^−1^ with CBP-HABP1. All peptides contained the same maximum in the amide II region (CN stretching and NH bending) at 1532 cm^−1^ and lacked peaks in the amide III region (CN stretching and NH bending) (Figure 5).

Next, we investigated the in vitro structural changes of the peptides in solution via circular dichroism spectroscopy (CD). We used the CDPro software package for CD spectral data processing and analysis [51]. Figure 6 shows the estimated changes in the secondary structure properties of the collagen binding (Figure 6A), HABP1 (Figure 6B), and chimeric peptide CBP-HABP1 (Figure 6C) [51]. The change in secondary structure composition of the chimeric peptide was compared to the mean secondary structure compositions of the single functional domains (Figure 6D). The chimeric peptide appeared to develop a more unordered structure compared to the single functional domains. Still, the compositional changes for all the features were observed to be 10% or less. We next evaluated the peptide functionality to determine if these changes contributed to the magnitude of change in secondary structure features which would have measurable impacts on the chimeric domain functions.

### 2.1. Effect of Peptide Incorporation on Collagen Self-Assembly

Building upon the conformational flexibility of the peptide, we next continued our experiments with the CBP-HABP1 peptide to study the assembly properties on the collagen fibrils on mica substrate using AFM topography image analysis (Figure 7). The images were acquired in tapping mode and presented as 5 × 5 μm images with the pseudo-color scale shown next to the image and the gradient change corresponding to observed surface features. Non-fibril features, indicated as intensity saturated regions on the surface, are acetic acid salt crystals from the evaporation of the solution in which the collagen was suspended. These features were excluded from image to prevent signal shadowing of the surface fibril assembly. Topographical analysis of the surfaces showed changes in the collagen fibril assembly with the incorporation of peptide in collagen prior to drop-casting (Figure 7). Collagen alone (Figure 7A) was observed to have an average fibril width of 98.89 nm, collagen+(CBP) showed a larger average fibril width of 121.4 nm, and collagen+(CBP-HABP1) exhibited an average of 82.37 nm (Table 1). However, with collagen+(HABP1) the average fibril width rose to 129.9 nm, but the standard deviation more than tripled compared to that of the other samples (Table 1). We see in Figure 7C a mixture of larger fibrils as compared to fibrils formed with collagen alone (Figure 7A) or compared to fibrils formed with collagen+(CBP) (Figure 7B), or collagen+(CBP-HABP1) (Figure 7D).

Unpaired two-tailed *t*-tests were performed (*p* < 0.05) comparing the observed fibril widths between collagen functionalized with CBP-HABP1 (Figure 7D) and the controls (Figure 7A–C). Collagen+(CBP-HABP1) fibrils were measured to be smaller than collagen-alone fibrils, resulting in a statistically significant difference. Though the average fibril width for CBP-HABP1 was observed to be less than that of the controls, it has a larger distribution of fibril sizes relative to the mean compared to collagen alone and collagen+(CBP) (Table 1), as observed in Figure 7D. A distinct difference in the fibril width properties was observed when CBP and CBP-HABP1 were co-assembled with the collagen fibrils as compared to HABP1. This potentially implies similar molecular interactions are taking place between the those peptides (CBP and CBP-HABP1) and the collagen fibrils.

The fibril assembly with HABP1 seen in Figure 7C shows the wider variation of collagen fibril widths compared to collagen+(CBP) (Figure 7B) or collagen+(CBP-HABP1) (Figure 7D). The fibril width standard deviations observed on the collagen+(CBP) and collagen+(CBP-HABP1) samples were observed to be relatively similar compared to collagen+(HABP1) (Table 1). This finding suggests that there is a similar molecular interaction occurring between the collagen and CBP or CBP-HABP1, which differs in the case of collagen and HABP1. Future studies can determine the effect of the hydrophobic/hydrophilic dynamics of these peptides on collagen fibril self-assembly [52]. Furthermore, it could be further explored if the molecular interactions are between the same collagen-contact residues for CBP and chimeric CBP-HABP1.

### 2.2. Mechanical Properties of Peptide-Incorporated Collagen Platforms Pre/Post Mineralization

Building on the observed effects of the chimeric peptide on the collagen, we next examined if there was a change on the mechanical properties of the collagen due to the peptide presence. Quantitative modulus mapping was executed via PeakForce-QNM AFM to observe the changes in the elastic modulus distribution on the surfaces. The elastic modulus was previously established to be derived by using the Derjaguin–Muller–Toropov (DMT) model, and is thus reported as the DMT modulus [53]. Figure 8A shows the topographical view of the surface with peptides incorporated as compared to the collagen as the control. For the modulus mapping of those surfaces, Figure 8B, collagen alone shows a uniform, average elastic modulus, which is lower than those co-assembled with peptides (Figure 8C). Comparatively, CBP or CBP-HABP1 incorporated into collagen resulted in values of 5.16 and 5.25 GPa, respectively. The relatively similar modulus values were observed to be much higher compared to collagen+(HABP1) or collagen control samples, shown in Figure 8C. Collagen+(CBP-HABP1) resulted in the largest increase in modulus compared to CBP or HABP1 peptides or collagen. The collagen+(CBP-HABP1) also showed a larger surface modulus variance, Figure 8B (Appendix A), compared to that of collagen. These results suggest that peptide-dependent nanomechanical enhancement is plausible to explain the induced change in moduli for collagen fibril assembly.

We next studied the peptide-guided remineralization on the collagen platform using ALP-mediated mineralization. As a result of mineralization, major topographical and morphological changes were observed, resulting in rougher surfaces based upon AFM analysis. Figure 9A provides the 3D topographical views. The topographical mineral characteristics mapped appeared to be significantly different between control and peptide-incorporated samples (Figure 9A). These differences between mineral formations are subsequently reflected in the modulus mapping, Figure 9B. The average moduli examined in Figure 9C show statistically significant changes between pre- and post-mineralization moduli per surface. Collagen alone showed the largest increase in elastic modulus, followed by collagen+(HABP1) but which also exhibited the largest standard deviation. The larger moduli of collagen and collagen+(HABP1) can be attributed to the potential supersaturation of mineral on the surface layer of collagen resulting in a non-uniform distribution of mineral deposition. This would lead to a greater variation in mineral formation at the substrate surface, as supported by the larger variance observed with HABP1 compared to the rest of peptide samples including collagen control (Figure 9, Appendix A). On the other hand, as a result of remineralization the elastic modulus of collagen+(CBP) and collagen+(CBP-HABP1) increased to 6.52 and 5.83 GPa, respectively, compared to the other samples. Although these values are lower than the modulus of HABP1 on collagen or control sample, the chimeric peptide (CBP-HABP1) had the smallest variations in mechanical properties, suggesting that it offered a more consistent route for directing peptide interactions within the collagen sample.

### 2.3. Examining the Ca-P Deposits on the Peptide-Incorporated Collagen Platform

The composition and morphology of the Ca-P deposits formed on collagen samples were examined using scanning electron microscopy (SEM) with energy dispersive X-ray analysis (EDS). Incorporation of CBP-HABP1 to collagen resulted in similar morphology compared to the HABP1 incorporation alone (Figure 10). The EDS results validated the finding of Ca-P mineral isomorphs on the peptide-incorporated collagen platforms. The calcium/phosphate ratios were calculated to examine compositional differences. The initial mineral formed in the collagen alone and collagen+(HABP1) platforms had an average Ca-P ratio of 1.40 and 1.41, respectively. Whereas collagen+(CBP) and collagen+(CBP-HABP1) had comparatively lower average Ca-P ratios at 1.30 and 1.25, respectively. These values are similar to the molar ratio of octacalcium phosphate, which is around 1.33, and amorphous calcium phosphate (ACP), which is roughly 1.50 [54,55,56,57]. The incorporation of the collagen-binding motif resulted in a decrease in the Ca-P ratio to 1.25 of the mineral deposits directed by CBP-HABP1. This ratio suggest the presence of brushite (DCPD) and OCP.

We also performed the spatial mapping of the mineral formed with or without peptide collagen samples (Figure 11). The surfaces are shown at 30,000× and processed using the Multi-Otsu thresholding algorithm, separating surface and subsurface mineral formation along with any background noise. Comparing mineral formations on SEM images, collagen and collagen+(CBP) both showed branched columnar mineral growth, whereas collagen+(HABP1) and collagen+(CBP-HABP1) showed more extensively branched columnar mineral growth as well as plate-like mineral growth.

Based on the Multi-Otsu calculations, the CBP-HABP1-incorporated collagen samples showed similar mineral area coverage compared to HABP1-incorporated collagen samples. The histogram distribution of the CBP-HABP1 was also like that of HABP1 on collagen samples, indicating a similar mineral growth amount at their respective locations, compared to the collagen+(CBP) or the control. This supports the preservation of HABP1 mineralization activity in the chimeric peptide. The results suggest a robust mineral formation taking place with the inclusion of the chimeric CBP-HABP1 peptide when incorporated on collagen samples.

We next examined the peptide-incorporated collagen samples following post-mineralization using the three-dimensional micro X-ray computed tomography (MicroXCT), shown in Figure 12. MicroXCT offers a non-destructive analysis of the internal structure with 3D imaging features and can be utilized as a physical indicator of intrafibrillar mineralization. The chimeric CBP-HABP1 peptide-incorporated collagen samples were examined post-mineralization and compared to the ALP-mineralized collagen samples. The shaded surface display (SSD) of the collagen control (Figure 12B) and the collagen+(CBP-HABP1) samples (Figure 12A) were examined to isolate the mineral structure on the platform from the surrounding collagen within the acquired volume set of the sample. The mineralized collagen layer incorporated with CBP-HABP1 (Figure 12A) showed denser formation compared to ALP-based mineralized collagen samples. The control collagen samples have clear voids areas distributed throughout the collagen interior regions, whereas CBP-HABP1-incorporated collagen revealed almost no void areas. 

## 3. Discussion

This study is the further investigation of our research on designing a self-strengthening peptide-functionalized dental adhesive to improve the integrity and durability of the adhesive/dentin (a/d) interface. Our previous work focused on improving the adhesive through the development of peptide-tethered-polymer systems [16] and self-strengthening adhesive formulations [1,15]. Here we designed an engineered peptide (CBP-HABP1) enabling collagen intrafibrillar mineralization to promote enclosure of demineralized collagen fibrils and help to improve the integrity and durability of the a/d interface.

The designed chimeric peptide, CBP-HABP1, exhibited conserved activity similar to its single domain collagen-binding motif on the collagen samples. AFM topography data showed similar fibril assembly with the incorporation of the CBP-HABP1 peptide compared to CBP—such topography was not observed with HABP1. The results with CBP-HABP1 and CBP can be attributed to the fact that both peptides contain the collagen-binding motif, thus likely bound to the collagen, in part, through sense–antisense domain interactions [58]. Whereas with HABP1, its non-specific interactions with the collagen fibrils may have attributed to the formation of larger fibril widths. Comparisons of dentin collagen fibrils formed in vivo were shown to have a diameter range of 80–100 nm that self-assemble in a hierarchical manner between second tier (microfibrillar) and third tier (fibrillar) levels [59,60]. Studying the assembly of the peptides on our collagen platform may offer a method to mimic such hierarchical progressions of fibril assembly and help to design effector molecules. Interestingly, the CBP-HABP1 peptide showed an average fibril diameter within the range observed for dentin [12].

The intermolecular interactions of collagen self-assembly mediated by CBP-HABP1 showed overall improvement in the nanomechanical properties of the collagen platform. While the CBP-HABP1 peptide aligns and molecularly interacts with the collagen, it also offers mineral deposition along these sites, via the hydroxyapatite-binding motif. The bifunctional peptide thus mimics tissue interfaces that promote intrafibrillar mineralization [52]. The intrinsically disordered feature of the peptide may provide greater flexibility in its secondary structure conformations, thus affecting the relative activity of the motifs. This platform provides a model of how the theoretical function achieved through conformation is characterized by the real functional output. Through spacer modification, this could be further investigated on this platform to optimize the function of both the collagen-binding and mineralization-related motifs in the chimeric peptide, naturally extending beyond to probing other motif combinations (e.g., antimicrobial [16]).

Early-stage mineral formation, prior to stages reaching octacalcium phosphate (molar Ca-P ratio, 1.33) has so far been minimally investigated. Here with the CBP-HABP1-integrated collagen platform, extensively branched columnar mineral growth as well as plate-like mineral morphologies were observed [60]. It is possible that a form of ACP is being observed when forming Ca-P minerals in aqueous environments, showing as low as 1.18 Ca-P molar ratios in early transient forms [57,61]. Mineral formations may also include DCPD as one of the possible apatite precursors in addition to ACP. In acidic environments, DCPD may undergo hydrolysis to more stable phases, and therefore it is widely utilized as bone cements. As Ca-P mineral chemical composition is strongly dependent on pH and surrounding calcium phosphate ion concentrations, further studies on peptide-mediated mineralization using this platform would benefit from examining different mineralization conditions while monitoring pH changes and separately under different pH conditions. Given the in vivo microenvironmental pressures created by the complex physiological environment and bacterial attack in demineralized dentin at the adhesive/dentin interface, it is crucial to investigate and further optimize the intended peptides ensuring optimal function [62,63,64,65,66,67].

Our findings support the bifunctional design and confirm the bifunctional activity of CBP-HABP1 for the respective motif functions using the collagen platform. The studies could be extended for further investigation of activity and efficacy of different engineered peptides, initial mineral formation, and intrafibrillar mineralization under different conditions that mimics the oral environment. While the results of this investigation are promising, the formulations have not been tested under conditions relevant to the oral cavity. Additional investigations are needed to analyze the behavior of these formulations under the chemical and mechanical stresses that occur during their function in the oral cavity.

## 4. Materials and Methods

### 4.1. Materials

Peptide synthesis required N-methyl morpholine (NMM), Wang amide resin, Fmoc-resin, Fmoc-amino acid building blocks, D-biotin, piperidine, and 2-(1H-benzotriazole-1-yl)-1,1,3,3-tetramethyluranium hexafluorophosphate (HBTU), which were purchased from AAPPTec LLC (Louisville, KY, USA). N, N-Dimethylformamide (DMF, 99.8%), trifluoroacetic acid (TFA, 99%), triisopropylsilane (98%), thioanisole (99%), and diethyl ether (99%) were obtained from Sigma-Aldrich (St. Louis, MO, USA). 1,2-ethanedithiol (95%), N,N Diisopropylethylamine (99.5%, nitrogen flushed), and hydroxymethyl (Tris) aminomethane hydrochloride (Tris-HCl, 99%+, extra pure) were purchased from Acros Organics (NJ, USA). Phenol (89%), calcium chloride dihydrate (99.7%), and sodium hydroxide (97%) were obtained from Fisher Scientific (Fair Lawn, NJ, USA). Glycerophosphate calcium salt was purchased from MP Biomedicals LLC (Solon, OH, USA) and 6N Hydrochloric acid solution from Fisher Chemical (Fair Lawn, NJ, USA). All chemicals in this study were used without further purification. We acquired FastAP Thermosensitive Alkaline Phosphatase from Thermo Fisher Scientific (Vilnius, Lithuania). Collagen Type 1 solution from rat tail (3 mg/mL, C3867, 95%) was purchased from Sigma-Aldrich (St. Louis, MO, USA). Highest grade V1 12 mm Mica discs and 12 mm atomic force microscopy specimen discs were obtained from Ted Pella Inc. (Redding, CA, USA). STKYDOT, an atomic force microscopy specimen disc adhesive, was purchased from Bruker Corporation (Camarillo, CA, USA).

### 4.2. Peptide Design

The selection of the collagenase–collagen-binding peptide (CBP) sequence, TKKLTLRT, was based as an analog of the original peptide sequence, TKKTLRT, reported by de Souza et al. (1992) [58]. The latter corresponds to the amino acid sequence derived from the nucleotide sequence of the complementary DNA strand coding for the human pro-α2(I) collagen domain, a molecule attacked by human fibroblast collagenase. de Souza et al. used the principle of hydropathic complementarity to develop a peptide sequence that would result in sense collagen domain to antisense peptide sequence binding given respective hydrophobic–hydrophilic residue interactions [68,69,70]. Employing that same principle, analogs to TKKTLRT were explored to identify hydropathic profiles that showed better alignment with respect to the amino acid sequence of the sense strand of the gene. From this, the sequence TKKLTLRT was chosen to be implemented. In addressing the biomineralization prong of this study, the hydroxyapatite-binding peptide (HABP1) sequence (MLPHHGA) was selected. The sequence was developed by Gungormus et al. (2008) [34] where it showed the highest binding affinity in directing the mineralization process of calcium phosphate (Ca-P). This study examines a novel bifunctional peptide, CBP-HABP1, by combining these two motifs, separated by a short and rigid spacer: TKKLTLRT-APA-MLPHHGA.

### 4.3. Peptide Synthesis

The CBP, HABP1, and CBP-HABP1 peptides were synthesized on the AAPPTEC Focus XC synthesizer (AAPPTec, Louisville, KY, USA), using a standard Fmoc solid-phase peptide synthesis protocol. The peptides were synthesized on Wang resin with the subsequent resin-bound peptides cleaved and their side chains deprotected, resulting in the canonical C-terminus functional group of carboxylic acid. The cleavage cocktail of CBP contains the following: TFA, phenol, triisopropylsilane, and water (90:5:2.5:2.5, *v*/*v* percent). For HABP1, the cleavage cocktail is as follows: TFA, thioanisole, ethanedithiol, triisopropylsilane, and water (87.5:5.0:2.5:2.5:2.5, *v*/*v* percent). Lastly, the cleavage cocktail for the bifunctional peptide CBP-HABP1 contains TFA, phenol, thioanisole, water, ethanedithiol, and triisopropylsilane (81.5:5.0:5.0:5.0:2.5:1.0, *v*/*v* percent). The peptides were left in cleavage cocktail on a rotator for 2 h, precipitated in cold ether, and then lyophilized.

Crude peptide purification was performed on a semi-preparative reversed-phase high pressure liquid chromatography (HPLC) Waters system, containing a Waters 600 controller and Waters 2487 Dual Absorbance Detector, using a 10 μm C-18 silica Luna column (250 × 10 mm, Phenomenex Inc., Torrance, CA, USA). The mobile phase is composed of phase A (94.5% HPLC grade water, 5% acetonitrile, 0.1% TFA) and phase B (100% acetonitrile). Lyophilized peptides were dissolved in 4 mL of phase A and purified at 0.5% phase B × min^−1^ on a linear gradient (5–85% phase B), performed at 3 mL × min^−1^, room temperature, with detection at 254 nm. Purified fractions collected were verified by the analytical Shimadzu HPLC system, composed of an LC-2010 HT liquid chromatograph and SPD-M20A prominence diode array detector, with a 5 μm C-18 silica Luna column (250 × 4.6 mm, Phenomenex Inc., Torrance, CA, USA). This mobile phase is composed of phase A (99.9% HPLC-grade water, 0.1% TFA) and phase B (100% acetonitrile) with the system run on a linear gradient with 1 mL × min^−1^ flow, 40 °C, detection at 254 nm. The purified peptides (CBP, HABP1, CBP-HABP1; see Appendix A, Appendix A, and Appendix A, respectively) were lyophilized and stored at −20 °C (Appendix A).

### 4.4. Fourier Transform-Infrared Spectroscopy (FT-IR)

The molecular structures of our peptides were verified using the PerkinElmer Frontier IR spectrometer with the universal attenuated total reflectance polarization accessory (Waltham, MA, USA). Lyophilized peptide was evaluated across the wavelength range of 4000 cm^−1^–550 cm^−1^ at a 4 cm^−1^ spectral resolution.

### 4.5. Circular Dichroism Spectroscopy (CD)

The secondary structure of each of the three peptides, CBP, HABP1, and CBP-HABP1 were measured through circular dichroism (CD) spectra. Measurements were made with CD spectrometer (JASCO, J-815) at room temperature, using a 1.0 mm cuvette. Each peptide sample was dissolved at 0.2 mg/mL in 10 mM potassium phosphate (pH 7.4) at 4 °C for 16 h. Spectra shown are averaged from three experimental repeats. The scans were acquired from 190 to 300 nm at a scanning speed of 60 nm/min. CD spectra were processed for secondary structure composition with the tools of CD Pro. For each peptide, the reference set selected was SMP50. The mean residue ellipticity (MRE) was analyzed with CD Pro software to compare likely conserved secondary structure features from the single domains (CBP and HABP1) in the chimeric peptide (CBP-HABP1), see Appendix A [51]. Absorbance measurements were taken every 1 nm as the average of 5 technical replicates and smoothed by a 7-point Savitzky–Golay filter. The fractions of secondary structure (Regular Helix, Distorted Helix, Regular Sheet, Distorted Sheet, Turns, and Unordered) were averaged for all three CD Pro tools (SELCON3, CDSSTR, and CONTILL).

### 4.6. Collagen-Peptide Sample Preparation

Grade V1 round mica, 0.21 mm thick and 12 mm in diameter, was used as the substrate surface and mounted onto a 12 mm atomic force microscopy specimen disc using STKYDOT adhesive. In 1 mL of 3 mg/mL Type 1 rat tail collagen suspension, 50 nanomoles of peptide were added. The ratio of mass collagen to mass peptide ranged from 100 wt collagen: 1 wt peptide for single domains to 100 wt collagen: 3 wt peptide for the chimeric peptide. Once peptide was dissolved and gently mixed in the collagen suspension, 150 μL was drop-cast onto the substrate surface and left to self-assemble at 4 °C for 16 h. The preparation resulted in mica surfaces with a coating layer of collagen-peptide, observed as “dry” by visual inspection.

### 4.7. ALP-Mediated Mineralization

ALP-driven mineralization was executed following the protocol discussed in Gungormus et al. (2008) [34]. The mineralization buffer prepared was composed of 24.4 Ca^2+^ mM and14.4 mM B-glycerophosphate made in 25 mM Tris-HCl buffer at pH 7.4, and 200 μL/well was added to a 24-well plate. Collagen-peptide samples were placed gently within the well to lay flat at the bottom. The mineralization reaction was initiated by adding FastAP (thermosensitive alkaline phosphatase), 1.4 × 10^−6^ g/mL, and left to incubate at 37 °C for 20 min. The samples were then removed from the wells, gently rinsed in sterile-filtered water, and left to air dry overnight. Throughout this process the samples were handled in a horizontally flat manner to prevent disturbing the peptide-functionalized collagen layer on the surface of the substrate.

### 4.8. RAMAN Spectroscopy

Demineralized dentin samples, adhesive/dentin samples (pre/post mineralization), and SC collagen on glass were imaged using a LabRAM ARAMIS Raman microscope (HORIBA Jobin Yvon, Edison, NJ, USA), equipped with a HeNe laser (λ = 663 nm, laser power = 17 mW). Light micrograph images were taken using a 50× long working distance objective Olympus lens. The samples spectra were evaluated over a wavelength range of 300 cm^−1^ to 1800 cm^−1^ with a 15 s spectra acquisition time of 4 acquisitions per cycle and processed using LabSPEC 6 software (HORIBA Jobin Yvon, Edison, NJ, USA). Divisive Clustering Analysis (DCA) was used to classify and group respective spectra. This multivariate analysis method was integrated into the LabSPEC 6 analysis software as a Multivariate Analysis (MVA) module (powered by Eigenvector Research Inc., HORIBA Jobin Yvon, NJ, USA). A rectangular area of the surface was imaged and submitted to this multivariate analysis where a statistical pattern determined derived independent clusters to present chemically distinct regions [3,4]. Average spectra are calculated per cluster which were used to provide information on particular peak parameters and component distribution in the resulting spectra. These components were analyzed for relative degree of mineralization via mineral-to-matrix ratio (MMR), inferred from the ratio of peak intensities at 960 cm^−1^ (phosphate) and 1460 cm^−1^ (CH_2_ bending), crystallinity via full width at half maximum (FWHM) from the ν1 phosphate band (960 cm^−1^), and the carbonate content of mineral crystallites via gradient in mineral content (GMC), based on ratio of relative carbonate (1070 cm^−1^) and phosphate (960 cm^−1^) peak heights. The calcium-phosphate (Ca/P) to collagen ratio was also assessed to understand the relative thickness of the mineral formed during mineralization [3]. This would help in comparative evaluations between the samples.

### 4.9. PeakForce-QNM AFM Imaging

A Bruker Multimode 8 HR scanning probe microscope (Bruker Nano Inc., Camarillo, CA, USA) was operated in peak force tapping mode with the capability of Quantitative Nanomechanics (PeakForce-QNM) in air mode conditions (24 ± 2 °C, 40% ± 5% RH). This advanced testing mode was used to examine the topographical and nanomechanical property changes of the collagen-peptide samples both before and after mineralization. Tapping mode etched silicon probes, type RTESPA 525-30 (Bruker Nano Inc., Camarillo, CA, USA) with a resonant frequency of about 518 kHz, were used to acquire images (1 μm × 3 μm and 5 μm × 5 μm) at scan rate of 0.5 Hz with 512 pixel/line resolution. The images of the samples were recorded using NanoScope 8.15 software and analyzed using NanoScope Analysis 2.0 software (Bruker Nano Inc., Camarillo, CA, USA).

### 4.10. SEM/EDX Imaging of Mineralized Collagen-Peptide Samples

In order to examine initial intrafibrillar mineral formation and morphology and determine Ca/P ratios precipitated on the peptide-functionalized collagen surface, a Cold Field Emission Scanning Confocal Microscope (SEM, S-4700 model, Hitachi High-Tech America, Schaumburg, IL, USA) equipped with a silicon drift energy-dispersive detector (EDS, X-max, Oxford Instruments, Concord, MA, USA) was used via the Microscopy and Analytical Imaging Research Resource Core Laboratory (RRID:SCR_021801). Collagen-peptide samples were sputter-coated with 3 nm gold using a Quorum sputter coating system (Q150, Quorum, Laughton, East Sussex, UK). SEM imaging was completed at an acceleration voltage of 10 kV at ultra-high resolution operating mode and EDS measurements were made at 10 kV under normal operating mode to preserve the integrity of the collagen-peptide layer and prevent burning during signal capture. EDS analysis was performed through AZtec software (X-MaxN, Oxford Instruments, Concord, MA, USA).

### 4.11. SEM Surface Mineral Mapping

SEM images were analyzed to examine the Ca-P mineral deposition changes between the different peptide-functionalized collagen platforms and to understand the relative amount of initial Ca-P mineral formed on those surfaces. The SEM images were processed through written code using Python 3.9.13 software (Appendix A). The tif files were preprocessed by median blurring and histogram equalization. Multi-Otsu threshold determination algorithm was applied, resulting in a binary matrix, with 1 corresponding to the location of mineral on the surface and 0 corresponding to a location of surface matrix (i.e. no mineral). The area of that represents the surface covered by mineral. The resulting binary image represents the surface mineral area, and the percent-area is the proportion of mineral surface locations to all locations in the image.

### 4.12. MicroXCT Imaging

The microscale structure of the mineral formed within the peptide-incorporated collagen platform as a physical indicator of intrafibrillar mineralization, was observed using 3D X-ray microcomputed tomography (MicroXCT-400, Xradia Inc., Pleasanton, CA, USA). The transmission X-ray images of the samples were obtained using a tungsten anode setting of 50 kV at 8 W. A total of 1600 images were acquired at a resolution time of 15 s per image. The 3D images were reconstructed using the XM Reconstructor 8.0 software and were analyzed from an orthogonal view using a TXM 3D Viewer under the spatial surface display (SSD) mode (Xradia Inc., Pleasanton, CA, USA).

## 5. Conclusions

Dental caries is the most ubiquitous infectious disease of mankind resulting in destruction of the teeth and is recognized as a global health crisis [54]. Composite resins are widely adapted in restorative dentistry, but their short lifespan leads to a cycle of repeated replacement along with an inherent risk of pulpal injury, loss of tooth structure, and weakened tooth resulting in fracture. The leading cause of composite restoration failure is recurrent decay. In contrast to dental amalgam, composite lacks the capability to seal gaps at the interface between the restorative material and tooth structure. The low-viscosity adhesive that bonds the composite to the tooth is intended to seal this interface, but the adhesive seal to dentin is readily damaged by chemical and mechanical stresses. The fragile dentin/adhesive bond would undoubtedly benefit from strategies that lead to enhanced durability and improved resistance to degradative substances including oral fluids, enzymes, and acids. The adhesive/dentin bond is a heterogeneous construct—its composition includes the hybrid layer, which ideally integrates adhesive with highly porous collagen fibers. Failure of the adhesive/dentin bond is the critical issue preventing long lasting dental restorations [4,10,71]. The weaker link resulting in recurrent decay is the hybrid layer, which integrates adhesive with the highly porous collagen fibers. Following acid-etching, demineralized collagen fibrils contain trapped water, which is extremely difficult to remove. Achieving complete enclosure of demineralized collagen fibrils by monomers to fill in and close those spaces is recognized as unattainable under clinical conditions. In nature, biomineralization adapted a progressive dehydration process where calcium phosphate precursors and proteins play a critical role in mineralizing the collagen interfibrillar domain, which results in excellent mechanical properties. We developed a bioinspired peptide-based approach enabling collagen intrafibrillar mineralization.

The chimeric peptide design incorporated collagen-binding and remineralization-mediating properties. We used a domain structure conservation approach in designing chimeric peptides with different spacer sequences and tested their structure–function relations. The selected chimeric peptide was tested on a type-I collagen-based platform. Peptides were studied for their self-assembly and remineralization properties on the collagen platforms. The engineered peptide was demonstrated to offer a promising route for collagen intrafibrillar remineralization.

Our future studies will build on recent collaborative work engineering advanced dental adhesive polymer networks by integrating multiple peptide functionalities [4,15,16]. We previously have shown antibacterial functionality with a methacrylate-conjugated antimicrobial peptide, and remineralization capability with a methacrylate-conjugated version of HABP1 [16]. Both peptide monomers can be polymerized into the same adhesive disc while maintaining their respective functions. In this study we engineered the peptide functionality across the interface. The proposed approach offers a platform to develop multifunctional strategies including different bioactive peptides, polymerizable peptide monomers, and adhesive formulations as a step towards improving long-term prospects of composite restorations.

## Figures and Tables

**Figure 1 ijms-24-06355-f001:**
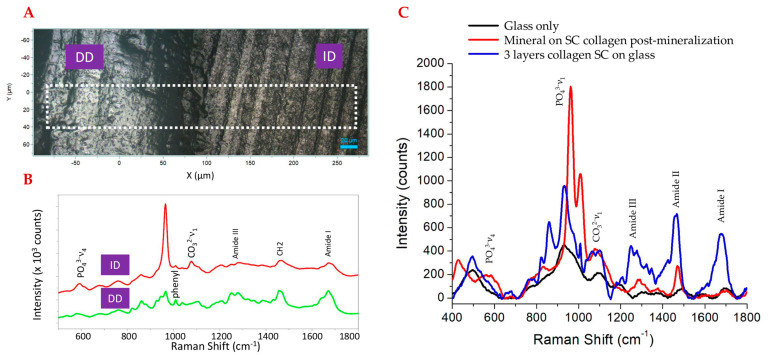
Human dentin profile. (**A**) RAMAN light microscopy image at 50× over 400 μm × 140 μm area, marked regions comparing demineralized dentin (DD) to intact dentin (ID). (**B**) RAMAN spectra of DD and ID highlighted regions. (**C**) RAMAN spectra of rat tail SC on glass pre/post 60 min of ALP−driven mineralization.

**Figure 2 ijms-24-06355-f002:**
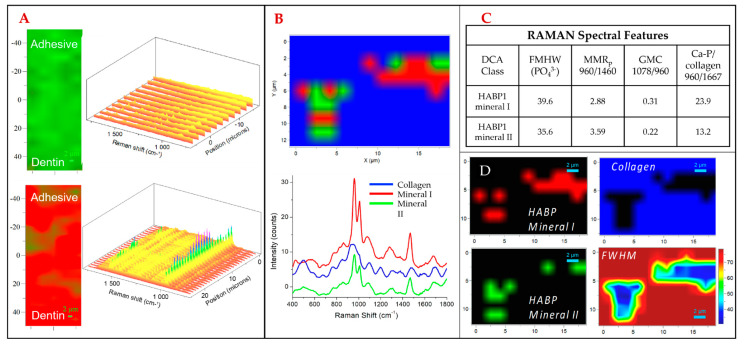
(**A**) RAMAN imaging with spectral analysis adjacent of A/D interface pre/post ALP−driven mineralization with HABP1 present. (**B**) RAMAN DCA of HABP1 mineralized surface on mineral types present and corresponding spectral analysis. (**C**) RAMAN spectral features evaluated over HABP1 mineral types observed. (**D**) Isolated DCA mapping of collagen, HABP1 mineral group I, HABP1 mineral group II, and full width at half maximum (FWHM) spatial mapping.

**Figure 3 ijms-24-06355-f003:**
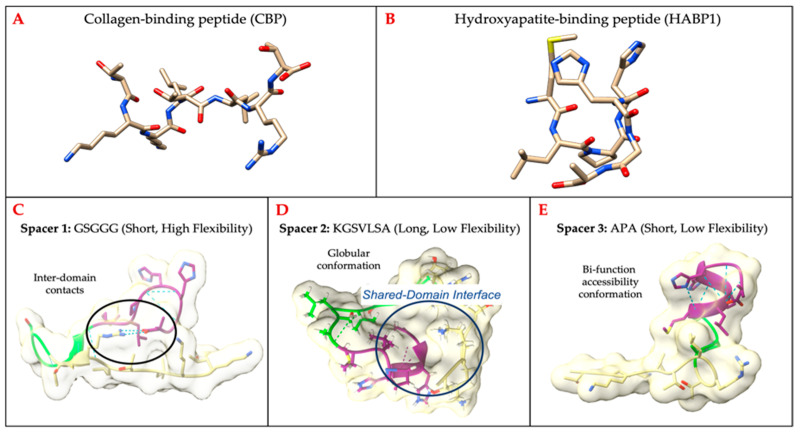
Folded peptide structures generated by PEP-FOLD3: (**A**) CBP and (**B**) HABP1. Building chimeric peptide CBP-HABP1: (**C**) short spacer with high flexibility, ex: GSGGG; (**D**) long spacer with low flexibility, ex: KGSVLSA; (**E**) short spacer with low flexibility, ex: APA.

**Figure 4 ijms-24-06355-f004:**
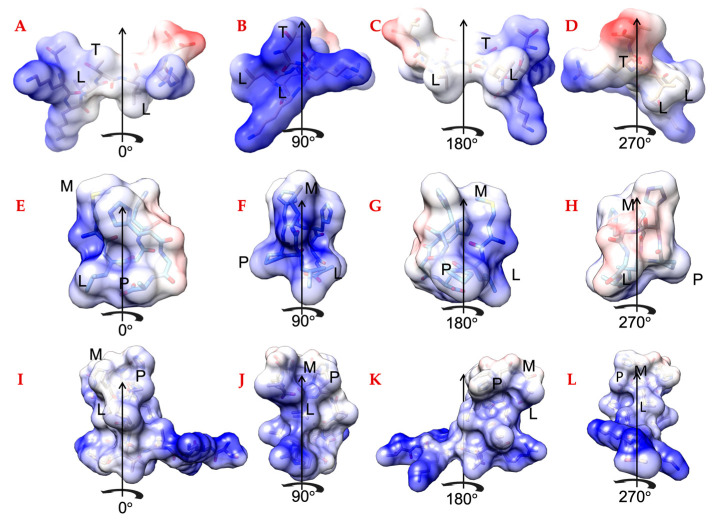
Electronegativity surface rotations of (**A**–**D**) Collagen-Binding Peptide (TKKLTLRT), (**E**–**H**) HABP1 (MLPHHGA), and (**I**–**L**) CBP-HABP1 (TKKLTLRT-APA-MLPHHGA). Underlined residues are labeled to orient the reader.

**Figure 5 ijms-24-06355-f005:**
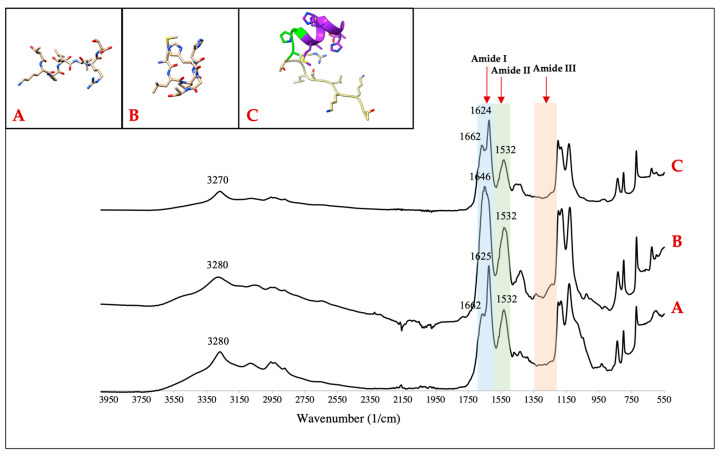
(**A**) Collagenase–collagen-binding peptide (CBP), sequence: TKKLTLRT. (**B**) Hydroxyapatite-binding peptide (HABP1), sequence: MLPHHGA. (**C**) Bifunctional peptide (CBP-HABP1), sequence: TKKLTLRT-APA-MLPHHGA. All peptide ribbon images shown are side views. All synthesized peptides are presented with corresponding FT-IR spectroscopy confirming molecular structure.

**Figure 6 ijms-24-06355-f006:**
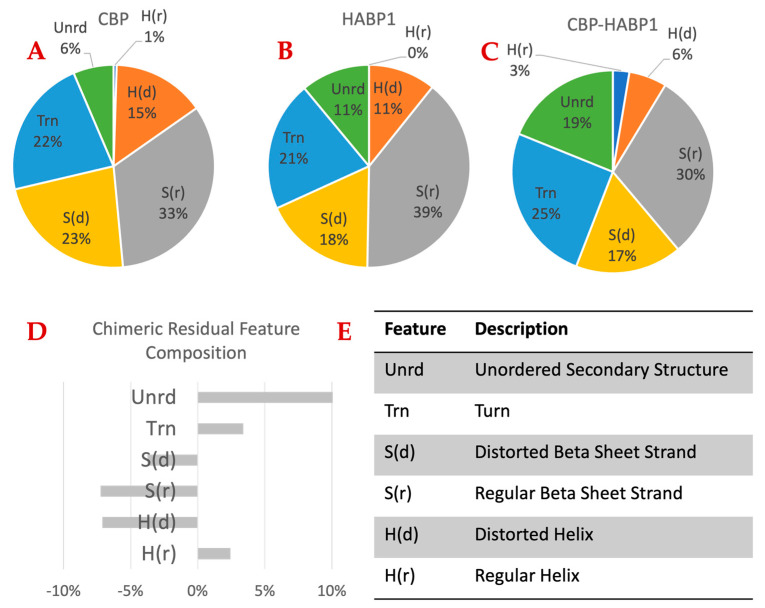
Circular dichroism additivity analysis comparing the average structure feature composition of among functional domains (**A**,**B**) to the chimeric peptide (**C**), which contains both functional domains and a 3-amino acid spacer. The residuals given in (**D**) are the difference between the feature composition from CD analysis and the average feature composition of the two domains. (**E**) Legend of feature descriptions in (**A**−**D**). The chimeric peptide has about 10% more unordered structure than expected from the average of the functional domains. This loss of ordered structure seems to come mainly from the loss of regular beta sheet strand (S(r)) and distorted helix (H(d)).

**Figure 7 ijms-24-06355-f007:**
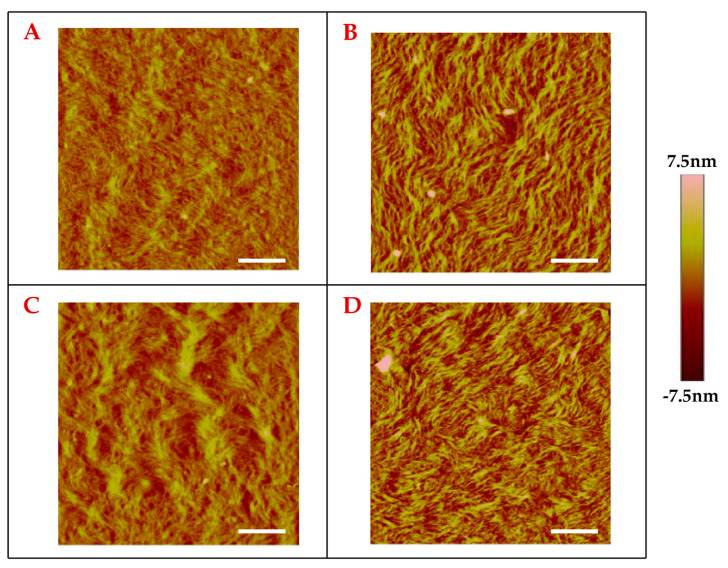
Collagen self-assembly examined through surface topography mapped via tapping mode AFM. Scale bar = 1 μm. (**A**) Collagen, (**B**) CBP on collagen, (**C**) HABP on collagen, (**D**) CBP−HABP1 on collagen.

**Figure 8 ijms-24-06355-f008:**
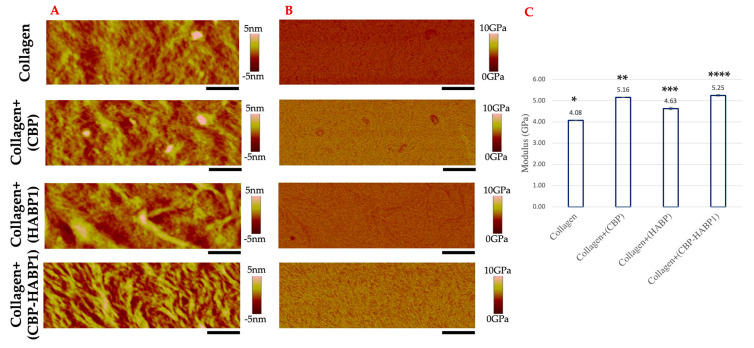
Mechanical properties of collagen control and peptide-functionalized collagen samples examined through PeakForce−QNM AFM. (**A**) AFM 2D-Topographical view. (**B**) DMT Modulus mapping. (**C**) Modulus averaged per pre-mineralization sample, error bars indicating standard deviation, and significant difference (*p* < 0.05) is represented as *, **, ***, ****. Scale bar = 500 nm.

**Figure 9 ijms-24-06355-f009:**
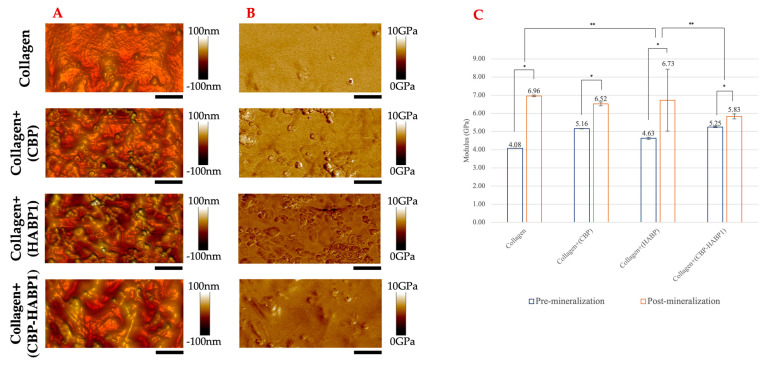
Characterization of peptide-functionalized collagen surfaces after 20 min of ALP-driven mineralization examined through PeakForce−QNM AFM. (**A**) AFM 3D-Topographical view. (**B**) Modulus mapping through the DMT modulus. (**C**) Comparison between pre- and post-mineralization per sample represented as averages over the area, error bars representing standard deviation, significant difference (*p* < 0.05) is represented as *, **. Scale bar = 1 μm.

**Figure 10 ijms-24-06355-f010:**
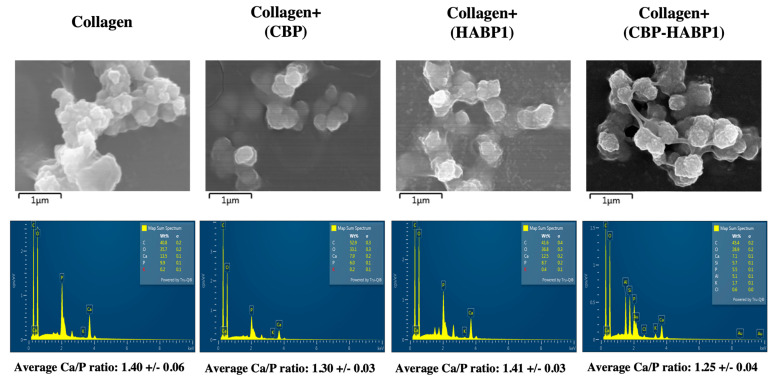
Mineral deposition on peptide-incorporated collagen samples via SEM images with corresponding EDS spectra. Ca/P averages displayed below EDS spectra, calculated across minimum three unique areas per peptide-functionalized collagen sample. Scale bar is under each image, 1 mm.

**Figure 11 ijms-24-06355-f011:**
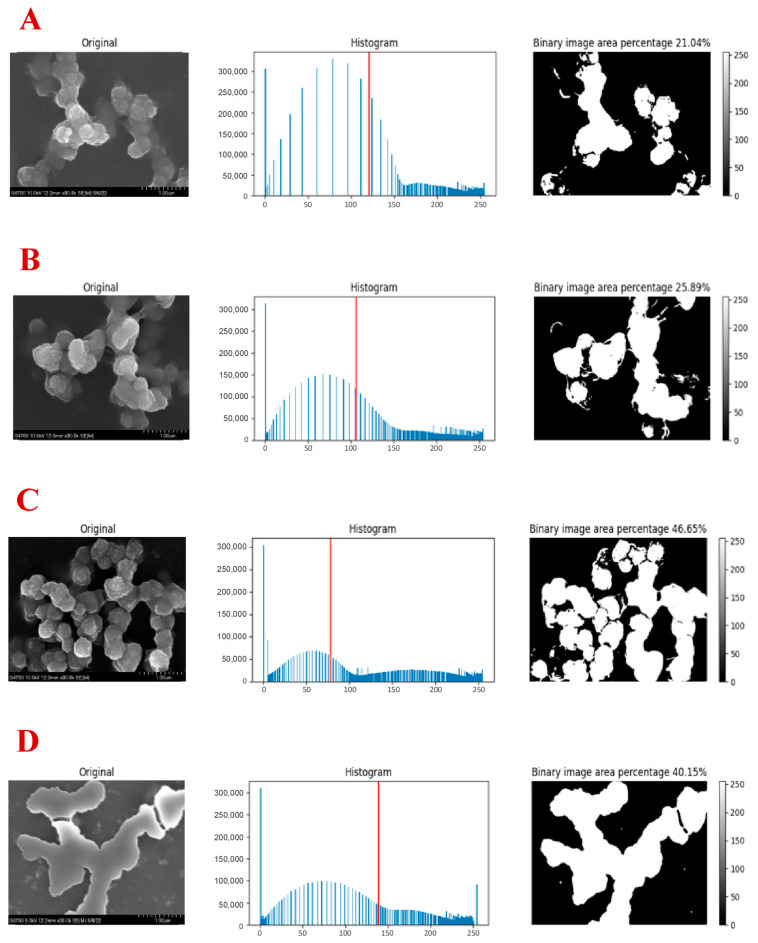
Mineral deposition mapping on SEM of peptide-functionalized collagen samples, magnification level 30,000×. The first image, in the 3 × 1 array per condition, is the original SEM image, the second is the histogram of the frequency of pixels per bin (0 to 255) within the SEM image, and the third is the binary conversion of the histogram based on the Multi-Otsu threshold and is indicated by the red line on the histogram. Gradient scale indicated on the right. (**A**) Drop-casted collagen; (**B**) Drop-casted collagen+(CBP); (**C**) Drop-casted collagen+(HABP1); (**D**) Drop-casted collagen+(CBP-HABP1). Solid orange arrows indicate branched columnar mineral growth and dashed cyan arrows show plate-like mineral growth.

**Figure 12 ijms-24-06355-f012:**
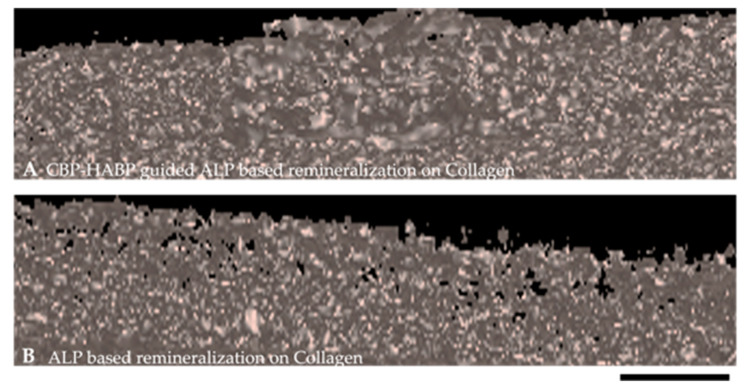
Shaded surface display of mineral formation on peptide-functionalized collagen samples, via Micro-CT, after 20-min ALP-driven mineralization, magnification level 20×. (**A**) CBP-HABP guided ALP-based mineralized collagen samples. (**B**) ALP-based remineralization on collagen samples. Scale bar is under each image, 50 μm.

**Table 1 ijms-24-06355-t001:** Statistics on fibril widths evaluated from AFM topographical imaging.

	Collagen	Collagen- (CBP)	Collagen-(HABP1)	Collagen- (CBP-HABP1)
Mean (nm)	98.89	121.40	129.90	82.37
Standard Deviation	19.26	20.66	87.643	26.47
Coefficient of variation	19.47	17.01	67.48	32.14

## Data Availability

Not applicable.

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
