# Peer review of "Engineered Peptides Enable Biomimetic Route for Collagen Intrafibrillar Mineralization"

_ijms, 2023, doi:10.3390/ijms24076355_

Round 1
Reviewer 1 Report
This is a clear and well-structured paper, elucidating the structural changes given by using different peptide clusters to improve the remineralization properties of the peptides on the collagen platforms. The mechanical properties of the collagen fibrils remineralized by the peptide assemblies were also adequately studied. The study is interesting on the methodological side and relevant to improve the strategies based on bioactive peptides. The limitation of the study have not been comprehensively addressed, in view of the translation of these findings into viable solutions for developing dental resins.
Reviewer 2 Report
The authors obtained and present data of the collagen assembly and mineral formation in the presence of a chimera peptide containing both the collagen assembly and the mineralization domains separated by a short link. Generally speaking, the results are well presented and the materials are well characterized by spectroscopic methods and by images. The secondary structure of the chimera peptides and of the separate domains were investigated by CD and infrared spectroscopy, but the energy-minimized folded structures may be interpreted as an approximation of the tertiary structure since no NMR data were presented.
Specific points:
Definitions of mineral groups I and II must appear clearly in the Results section.
Line 69 : Revise the grammar of the sentence starting in this line.
Line 111. Write “could be critical” instead of “could a critical”
In line 257, please write “circular dichroism spectroscopy (CD).”
In line 138 please write “reflecting the spectral analysis, respectively”
Line 153: Write “1454-1460 cm-1”
Line 225: I suggest the authors to use “Conversely,” instead of “Whereas”.
Line 283. Please check the significant figures of the fibrils widths reported.
Line 301: In the sentence “A distinct difference in the fibril width properties was observed when CBP and CBP-HABP1 are applied onto collagen fibrils, implying similar molecular interactions taking place among collagen and these peptides” the reported experimental observation is not related to the implication. Please, clarify.
Table 1: Change the title of the third line to “coefficient of variation”.
Line 310: Please write “which differs in case of collagen and HABP1” instead of “which lacks in case of collagen and HABP1”.
Line 311: The sentence “Larger standard deviations observed in case of HABP.” is grammatically incomplete.
Line 357: Write “CBP-HABP1”. Please, check this abbreviation throughout the text.
Line 382 (referring to Figure 10). “collagen+(HABP1) and collagen+(CBP-HABP1) show more extensively branched columnar mineral growth as well as plate-like mineral growth.” I am not able to observe either the branched columnar growth or the plate-like growth of the mineral structure, especially for collagen (CPB-HABP1). Could the authors please add more information to guide the eyes of readers that are less familiar with this methodology?
Lines 434-441. I suggest the authors for reviewing those sentences concerning the use of the English language.
Line 442. Are not those interactions intermolecular instead of intramolecular?
Line 462. Write “In acidic environments, DCPD may undergo” instead of “Acidic environments, DCPD may undergo”.
Line 509. I did not understand the meaning of “polymerizable peptide monomers” since there are not peptides that form polymers in the present manuscript.
Line 516. According to IUPAC recommendations, use small capital letters for the stereochemical notation: D-biotin.
Line 555. “side chains”.
Line 556 and in similar cases: “The cleavage cocktail of CBP contains the following: TFA, phenol, triisopropylsilane, water (90 : 5 : 2.5 : 2.5)”. Specify that the ratios are volume ones.
Line 563. According to IUPAC, write “reversed-phase”.
Section 4.3 Peptide Synthesis. How were the identity and the purity of the peptide checked? Mass spectra of the purified peptides must be furnished as supplementary material.
Section 4.5 Circular Dichroism Spectroscopy (CD). The authors must furnished the CD spectra with the molar ellipticity plotted as function of wavelength. In this section, please inform how the molar ellipticity was calculated.
Line 596. Write “50 nanomoles of peptide were added”.
Line 289. (Figure 7D) instead of (Figure 7C).
Figure 6. The circular dichroism spectra must be included as supplementary material.
Figure 8: In the legend write “Modulus averaged”. In Fig 8c, the asterisk symbol (*) is used for all the bars, which indicates that there is no significant difference between the modulus values. Please, address this point.
Figure 9. Add legends to the two lower images of figures 9A and 9B.
